# Interpersonal brain synchronization during sensorimotor synchronization in people with different aerobic fitness levels: A fNIRS-based hyperscanning study

Zixin Wang[1], Chengliang Wang[1], Shilin Wen[1]*, Haoteng Yuan[2], Shengyu Dai[3], Jianguang Cai[4], Liang Jiang[5], Xue Chen[5], Xianzhi Gao[5], Hui Xu[6]

**1** Capital University of Physical Education And Sports, Beijing, China, **2** Guangzhou Huashang Vocational College, Guangdong, China, **3** Fujian Normal University, Fujian, Fujian, China, **4** Hunan University of Science & Technology, Hunan, China, **5** Beijing Institute of Technology, Beijing, China, **6** Army Engineering University of PLA, Jiangsu, China

* 313811819@qq.com

## Abstract

Sensorimotor synchronization (SMS) refers to the temporal coordination of individual actions with perceptible external events and rhythms. Previous research has revealed a potential relationship between SMS abilities and physical activity, with proposed links to underlying inner-brain cognitive processes. However, it needs to be explored that whether good aerobic fitness will have a strong SMS ability, its internal mechanism, and the inter-brain mechanism. In the present study, we recruited 23 dyads of long-distance runners as the experimental group and 22 dyads of non-physical education major students as the control group. Participants were required to perform a dyadic finger-tapping synchronization task. During task performance, neural activities were recorded in frontal area by the functional near-infrared spectroscopy (fNIRS)-based hyperscanning approach. The results revealed that the experimental group demonstrated superior SMS abilities in the seconds-scale task. This advantage was attributed to their enhanced stability. Importantly, significant interpersonal brain synchronization(IBS) in the right superior frontal gyrus (rSFG) was observed only in the experimental group during SMS task performance, and this IBS was associated with their superior stability. These findings provide new evidence supporting the relationship between physical activity, cognitive capabilities, and associated neural plasticity.

## Introduction

Sensorimotor synchronization (SMS), a fundamental and crucial cognitive ability, refers to the temporal coordination of individual actions with perceptible external events and rhythms [1–3]. Achieving precise SMS requires individuals to accurately time their responses to external stimuli, which is a key factor in enhancing motor skills [4].

**Data availability statement:** All relevant data are within the manuscript and its Supporting Information files.

**Funding:** This research was funded by Humanities and Social Science Fund of Ministry of Education of China (22YJA890028). The funders were involved in the experimental design, data analysis, and the writing and revision of the manuscript.

**Competing interests:** The authors declare no conflicts of interest.

**Abbreviations:** SMS: sensorimotor synchronization; fNIRS: functional near-infrared spectroscopy; IBS: interpersonal brain synchronization; rSFG: right superior frontal gyrus; SMT: synchronized metronome training; IOI: interonset interval; ITI: intertap interval; PCG: phase correction gain; HbO: oxyhemoglobin; HbR: deoxyhemoglobin; HbT: total hemoglobin; ROI: region of interest; CHs: channels; MITI: mean intertap interval; MA: mean asynchrony; ITIV: intertap interval variability.

Aerobic fitness refers to the capacity to sustain muscular power or speed over extended periods, and is also commonly termed cardiovascular fitness [5]. It is well established that superior aerobic fitness can be achieved through regular physical exercise. A substantial body of research has demonstrated that consistent engagement in physical activity, or possessing a high level of aerobic fitness, not only contributes to overall physical health but also increases the production of beneficial neurochemicals during exercise. This, in turn, enhances neural circuit efficiency [6,7], induces structural and functional changes in the brain, and yields long-lasting, positive effects on a range of cognitive functions [8–12]. Evidence from neuroimaging studies indicates that a key brain region involved in these effects is the prefrontal cortex. For example, studies have found that aerobic exercise can increase the volume and density of the prefrontal cortex, thereby enhancing memory function [13]. In another study, children aged 8–9 years engaged in one hour of physical activity daily after school for nine months [14]. Functional MRI was used to monitor cerebral blood flow during a flanker task that assessed attention and inhibitory control. Results revealed that these children exhibited behavioral performance and prefrontal activation patterns comparable to those of college students. In the context of SMS tasks, the prefrontal cortex operates as part of the "cognitively controlled timing" system and plays a critical role in task execution [15,16]. However, it remains unclear whether optimal aerobic fitness can enhance SMS abilities, and the underlying neural mechanisms require further investigation.

In laboratory research on SMS, information processing methods are commonly employed, with the finger-tapping being a typical task where individuals strive to synchronize their finger taps with an external rhythm [1,17]. It is found that the performance of SMS tasks is directly influenced by the sequence interonset interval (IOI), with the optimal synchronization frequency occurring at approximately 2 Hz (around 400 ms-600 ms) [18]. At this frequency, performance differences among different populations are relatively small. However, with IOI extending, task performance declined and difference among individual increased [19].The forward prediction model offers a plausible explanation for these findings [1]. According to this model, individuals generate predictions in advance regarding the sensory consequences of their actions and subsequently compare these predictions with actual sensory feedback to make corrections—this forms the so-called "prediction– execution– feedback– adjustment" loop. The key components of this loop are prediction and adjustment. As the IOI increases, the difficulty in generating accurate predictions becomes substantially greater, thereby leading to increased individual variability. Aerobic exercise has been shown to promote structural and functional plasticity in brain regions intimately involved in motor control, such as the prefrontal cortex. This enhancement boosts the efficiency of information integration and processing within these regions, thereby strengthening the predictive and adaptive elements of the aforementioned loop [13]. Furthermore, in line with dynamic systems theory, aerobic exercise augments the dynamic regulation of the perception– action– environment coupling [20]. Thus, we hypothesize that individuals with higher levels of aerobic fitness may demonstrate superior performance in SMS tasks.

According to information processing approach, two crucial abilities are involved in the processing of SMS tasks. The first is the correction capacity for the intertap

interval(ITI) [1]. Phase correction serves as the key error correction processes in isochronous SMS tasks. It entails adjusting the local phase of the biological timer based on the degree of asynchrony observed in the most recent interaction with the external rhythm when there are no changes in the external rhythm. The degree of correction is commonly represented by the phase correction gain(PCG), which reflects the proportionate relationship between the correction time and the most recent asynchrony [21]. Research has indicated that during successful performance of SMS tasks, PCG typically ranges between 0.2 and 0.8 [22]. Furthermore, PCG tends to increase as the IOI lengthens [23]. The second is the capacity to maintain stability in the ITI. Even in the most elementary SMS tasks, biological timing continuously undergoes changes as the task progresses, resulting in sustained deviations from the target ITI [24]. These deviations also tend to increase with longer IOI. Exceptional stability capacity serves to diminish the magnitude of these deviations [25].

Inner-brain mechanisms underlying individual SMS have been identified that the functional brain regions involved in the task process and the activated brain regions corresponding to different task components [26]. However, with the advancement of cognitive neuroscience, it has been recognized that inter-brain coupling also exists during interactive processes [27]. Hyperscanning refers to the simultaneous recording of inter-brain activity among two or more individuals engaged in social interaction [28]. By analyzing the relationship between inter-brain activity and behavioral indicators, hyperscanning aims to reveal the inter-brain mechanisms underlying social interaction. Since its inception, the interpersonal brain synchronization(IBS) activities have become a research focal point [29]. Multiple participants to walk together in synchronized rhythm revealing that IBS are significant enhancement in the prefrontal regions [30]. It has been demonstrated that a positive correlation between IBS in regions such as the sensorimotor cortex and the degree of gesture synchronization on action synchronization [31,32]. These results suggested that the IBS in the prefrontal regions may serve as a crucial neural marker for SMS. Meanwhile, hyperscanning studies related to physical activities have found that both cycling and basketball are associated with enhanced IBS in the prefrontal cortex, as well as improvements in specific task performance [33,34].

Based on the beneficial effects of aerobic exercise on cognitive function and the enhancement of IBS in the prefrontal cortex induced by physical activity, the present study proposes the following hypothesis: Individuals with higher levels of aerobic fitness may exhibit superior SMS abilities, an advantage that becomes more pronounced as the IOI increases, possibly due to aerobic exercise enhancing task performance by facilitating IBS in the prefrontal cortex.

## Materials and methods

### Participants

The experiment was conducted between 11/ 11/ 2021 and 1/ 12/ 2021. A total of 100 male college students were recruited from a university in Hunan, China, including 50 members from the long-distance running team and 50 non-physical education majors. The long-distance running team members were assigned to the experimental group, while the non-physical education major students were assigned to the control group. All participants self-reported as right-handed. None of the participants had received systematic musical instrument training. Members of the experimental group engaged in an average of no less than 6 hours of aerobic exercise per week in the past three months, while members of the control group self-reported an average of no more than 1 hour of exercise per week during the same period and had no history of systematic gym workouts. To control for the influence of familiarity, participants who had known each other for more than 3 months were paired as matched pairs, resulting in a total of 50 pairs of participants [34].

One week prior to the experiment, the participants' 1000-meter running times were recorded as a measure of their aerobic fitness level [35,36]. During the data analysis process, it was discovered that 5 pairs of participants had abnormal behavioral or neural data, and they were subsequently excluded from the analysis. This resulted in a final sample of 45 pairs of participants, with 23 pairs in the experimental group (mean age: $21.20 \pm 1.98$ years) and 22 pairs in the control group (mean age: $20.32 \pm 1.60$ years). An independent samples t-test was conducted to compare the differences in 1000-meter running times between the two groups, revealing that the experimental group had significantly lower times compared to the control group (experimental group: $202.06 \pm 11.03$ s; control group: $239.12 \pm 13.02$ s; $t = -15.358$, $p < 0.001$).

Prior to the start of the experiment, each participant provided informed consent and had a clear understanding of the experimental procedures and content. This study received approval from the Institutional Review Board of the Capital University of Physical Education and Sports.

## Ethics statement

This study was performed in accordance with the ethical standards specified in the latest Helsinki Declaration and was approved by the the ethics committee of Capital University of Physical Education And Sports (2021A21). All participants signed the informed consent before the experiment and received certain remuneration after completing the experimental task.

## Procedures and tasks

After recording the participants' information, they were instructed to sit back-to-back in front of computer screens and read through the experimental procedures. Once the participants fully understood the instructions, they completed at least one practice trial for each condition before the formal experiment began (Fig 1A).

The experiment consisted of two main period: a resting period and a task period. The resting period lasted 2 minutes, after which the participants entered the task period. The task involved an auditory-paced synchronization task with three conditions(IOI: 500 ms, 1000 ms, and 1500 ms). Each condition consisted of 6 trials, resulting in a total of 18 trials. Each trial lasted 60 seconds, with the first 10 seconds presenting a hunt of metronome (low-pitched pure tone "beep") in the participants' headphones, during which they were instructed to synchronize their tapping with the system's rhythm. The remaining 50 seconds presented the partner's tapping sounds, and the participants were asked to maintain synchronization with both the system's rhythm and their partner's tapping.

The experimental trials were presented in a randomized order. After completing all 6 trials of one IOI condition, the participants moved on to the next IOI condition. There was a 30-second break between the different IOI conditions. The entire experiment lasted approximately 28 minutes (Fig 1B).

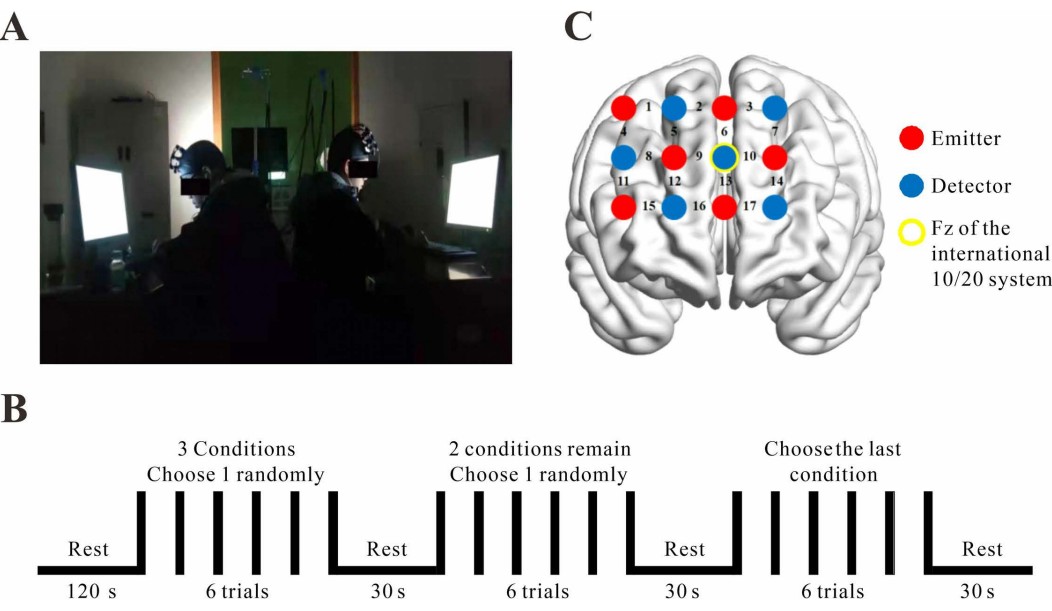

**Fig 1. Experimental design.** (A) Experimental scene. (B) Task procedure. (C) Optode probe set.

## Data collection

The experiment was conducted in a dimly lit psychological laboratory. The experimental stimuli were programmed using MATLAB(2018a) and the Psychtoolbox toolbox(3.0). The display had a screen resolution of 1920 × 1080 pixels and a refresh rate of 60 Hz. Auditory stimuli were presented through earphones, with the volume set to a comfortable level determined by the experimenter.

Participants performed the tapping task using a keyboard, and their tapping time points were recorded and saved using MATLAB. An FOIRE-3000 optical topography system (LABNIRS; Shimadzu Corporation, Japan) was used to record the oxyhemoglobin (HbO), deoxyhemoglobin (HbR) and total hemoglobin (HbT) concentrations for each dyad. The absorption of near infrared light (wavelengths: 790, 830 and 830 nm) was measured at a sampling rate of 10 Hz. The prefrontal cortex was selected as the region of interest (ROI) for this study. The optode probe set was placed over participants' prefrontal cortex (Fig 1C). Two 3 × 4 optode probe sets (six emitters and six detectors, 3 cm optode separation) were used. Each set consists of 17 measurement channels (CHs). The second row's third optode was placed at Fz, according to the 10/20 international system. The virtual registration method was used to determine the correspondence between the NIRS CHs and the measured points on the cerebral cortex (Table 1) [37].

## Data processing and analysis

### Behavioral data preprocessing.

1. The time points of the participant's taps in each trial were extracted using Matlab.

2. The first 10 seconds of tapping data in each trial were removed.

3. If a participant pressed the key twice within 0.2 seconds, the second key press was deleted.

**Table 1. The coordinates in MNI space and corresponding neuroanatomical labels for channels in the ROI.**

| Channel | MNI coordinates (E/C) | | | Brodman area |
|---|---|---|---|---|
| | X | Y | Z | |
| 1 | 27/28 | 25/24 | 61/62 | Dorsolateral prefrontal cortex |
| 2 | 5/6 | 27/26 | 64/65 | Dorsolateral prefrontal cortex |
| 3 | −17/-17 | 28/28 | 63/63 | Dorsolateral prefrontal cortex |
| 4 | 41/40 | 32/31 | 48/48 | Dorsolateral prefrontal cortex |
| 5 | 18/19 | 36/36 | 58/58 | Dorsolateral prefrontal cortex |
| 6 | −7/-6 | 40/40 | 57/57 | Dorsolateral prefrontal cortex |
| 7 | −26/-26 | 36/36 | 53/51 | Dorsolateral prefrontal cortex |
| 8 | 30/31 | 44/44 | 45/44 | Dorsolateral prefrontal cortex |
| 9 | 10/11 | 48/49 | 52/51 | Pre-Motor and Supplementary Motor Cortex |
| 10 | −14/-13 | 48/49 | 49/49 | Includes Frontal eye fields |
| 11 | 46/46 | 49/48 | 26/27 | Includes Frontal eye fields |
| 12 | 23/24 | 56/56 | 38/39 | Pre-Motor and Supplementary Motor Cortex |
| 13 | 0/0 | 56/56 | 39/39 | Pre-Motor and Supplementary Motor Cortex |
| 14 | −23/-23 | 56/57 | 36/36 | Pre-Motor and Supplementary Motor Cortex |
| 15 | 38/37 | 60/60 | 20/21 | Pre-Motor and Supplementary Motor Cortex |
| 16 | 14/15 | 67/66 | 29/28 | Includes Frontal eye fields |
| 17 | −10/-10 | 66/66 | 28/28 | Includes Frontal eye fields |

Note: E represents the experimental group, and C represents the control group.

4. Within the same trial, if a participant missed a key press, the missing value was imputed using the average of the adjacent key press time points. If three or more key presses were missed, the entire trial was excluded.

5. If the number of key presses differed between participants within the same trial, the data were selectively trimmed (either the first or last key press) to ensure consistent tapping counts across participants [38].

After excluding trials that did not meet the processing requirements, 741 trials remained. Among them, there were 251 trials with a IOI of 500 ms, 252 trials with a IOI of 1000 ms, and 238 trials with a IOI of 1500 ms. These remaining 741 trials were included in the formal analysis.

## Behavioral data processing and analysis

Mean intertap interval(MITI): The MITI is used to assess whether the two groups of participants performed the task under the same conditions. It is obtained by calculating the mean interval between key presses for each participant within each trial. Previous research has found a strong correlation between the MITI and task performance. If there is a significant difference between the two groups, it indicates inconsistent task conditions between the groups, which can directly impact other experimental results [1].

Synchrony Performance: Mean Asynchrony(MA) is used to evaluate the performance of the SMS task. It is obtained by subtracting the key press time point of one participant from the corresponding key press time point of another participant, and then calculating the mean of the absolute values of the asynchronies. In the same condition types, a smaller MA represents stronger synchrony ability.

Correction Ability: The PCG is selected as an indicator to evaluate the correction ability. PCG represents the extent to which participants adjust the timing of their next key press based on the previous asynchrony. The closer the average value of PCG is to 1 for each pair of participants [38], the stronger the correction ability of the synchrony system. The calculation method adopts the bivariate Generalized Least Squares (bGLS) model [39].

Stability: intertap interval variability(ITIV) is used as an indicator to assess stability. In the same condition types, a smaller ITIV represents more stable time control ability. It is obtained by calculating the standard deviation of the ITI for each participant within each trial.

Independent sample t-tests were performed using SPSS 24.0 statistical software to analyze the MITI. A repeated measures analysis of variance (ANOVA) with two groups (experimental group and control group) and three conditions (IOI: 500 ms, 1000 ms, 1500 ms) was conducted for the remaining behavioral data. To determine the main strategies used in task completion, Pearson correlation analysis was employed to examine the relationship between MA, PCG and ITIV.

## fNIRS data analysis

Previous studies have found that HbO is more sensitive to changes in blood flow [40,41]. Therefore, in this experiment, only HbO signals were selected for analysis. Firstly, trials that were excluded during the behavioral data analysis were removed. To eliminate interference from environmental changes on the signals, the first 20 seconds of the resting period were excluded, and the remaining HbO signals were included in the analysis. Resting period brain data from the last 100 seconds (excluding the first 20 seconds) were selected as the baseline for each participant and each recording channel. The data from 50 seconds following the end of the auditory stimulus in each trial were selected as the task period data. Next, we used a principal component spatial filter algorithm (i.e., PCA approach using Gaussian spatial filtering) to remove the global components [42]. Wavelet transform coherence (WTC) was then used to assess the relationship between HbO time series for each dyad [43]. The results showed that our task was more sensitive to the frequency band between 25.6 seconds and 51.2 seconds (frequency range: 0.039 Hz – 0.020 Hz) (Fig 2). In this experiment, each trial lasted for 50 seconds (0.02 Hz), so the corresponding frequency band range selected was from 0.020 Hz (51.2 s) to 0.039 Hz (25.6 s).

 

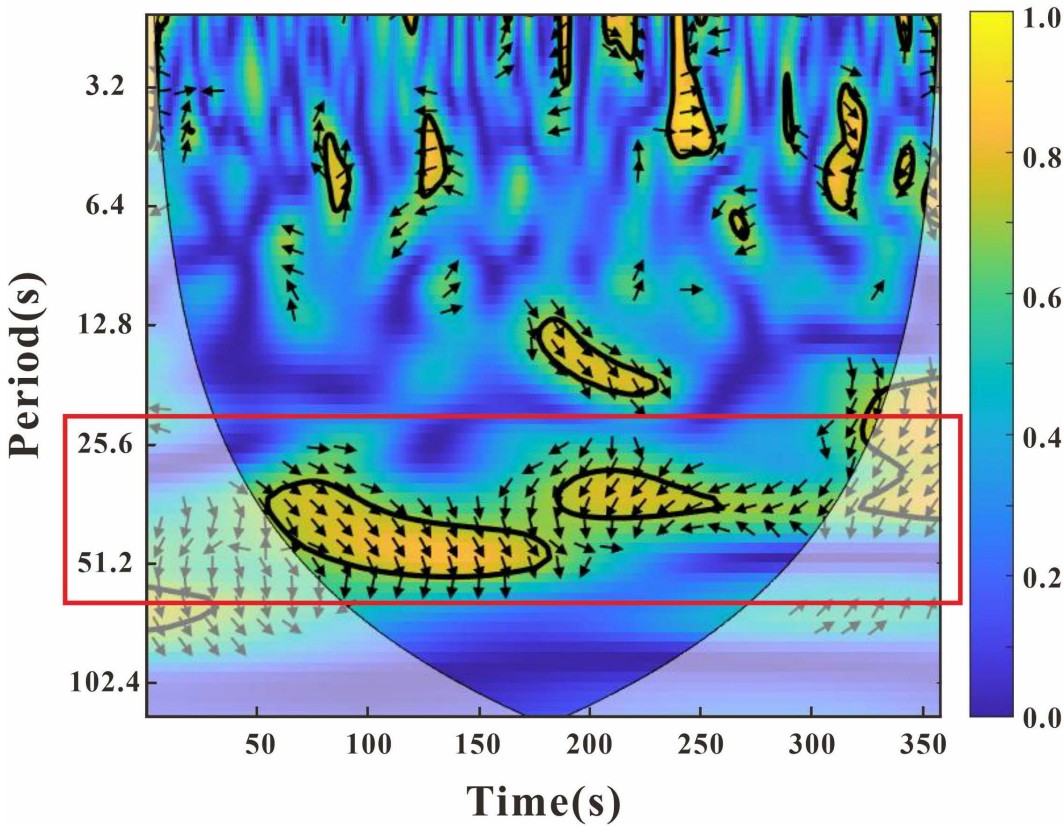

**Fig 2. Frequency band of interest.** Interpersonal brain synchronization (IBS) indicated by wavelet transform coherence (WTC). The coherence based on HbO signal from channel 2 (CH2) in a representative pair of experimental group. The red border represents the frequency band of interest (25.6 s −51.2 s), indicating when the task was carried out. The color bar denotes the value of WTC(1 = highest coherence, 0 = lowest coherence).

For each pair of CHs between two participants, the average interpersonal brain synchronization enhancement, or IBS, was calculated in task period and resting period. We defined IBS as the task-related interpersonal brain synchronization enhancement (i.e., mean(block) – baseline). The larger the value, the stronger the task-related enhancement of IBS. Then, we converted IBS values to Fisher z-statistics and performed a one-sample t-test against zero across all channels, applying false discovery rate (FDR) correction ($p < 0.05$).

For channels showing significant IBS, a repeated measures analysis of variance (ANOVA) with two groups (experimental group and control group) and three conditions (IOI:500 ms, 1000 ms, 1500 ms) was conducted. Pearson correlation analysis was used to examine the relationship between the IBS and behavioral performance.

## Results

### Behavioral results

**Mean intertap interval(MITI).** The MITI for the experimental group were 494.19 ± 19.48 ms, 961.94 ± 79.62 ms, and 1455.48 ± 161.07 ms (IOI: 500, 1000 and 1500 ms, respectively), while for the control group, they were 503.84 ± 28.98 ms, 985.02 ± 80.37 ms, and 1438.16 ± 177.50 ms (IOI: 500, 1000 and 1500 ms, respectively). Independent samples t-tests revealed no significant differences between the two groups [500 ms IOI: $t_{(43)} = -1.316$, $p = 0.195$, Cohen's d = −0.391; 1000 ms IOI: $t_{(43)} = -0.968$, $p = 0.339$, Cohen's d = −0.289; 1500 ms IOI: $t_{(43)} = 0.343$, $p = 0.733$, Cohen's d = 0.102]

(Fig 3A).This result indicates that both groups were under consistent task conditions across the three tasks, which adds credibility to the analysis of the task performance indicators.

## Synchronization performance - Mean Asynchrony (MA)

The MA for the experimental group was 57.82 ± 12.10 ms, 79.50 ± 19.61 ms and 119.04 ± 22.25 ms (IOI: 500, 1000 and 1500 ms, respectively), while for the control group, it was 62.92 ± 12.20 ms, 91.69 ± 19.86 ms and 150.44 ± 20.50 ms (IOI: 500, 1000 and 1500 ms, respectively). Repeated measures ANOVA revealed a significant main effect of group [$F_{(1,43)}$ = 18.066, $p < 0.001$, $\eta^2 = 0.296$], a significant main effect of condition [$F_{(2,86)}$ = 255.432, $p < 0.001$, $\eta^2 = 0.856$], and a significant interaction effect of group × condition [$F_{(2,86)}$ = 8.271, $p = 0.001$, $\eta^2 = 0.161$]. Further simple effect analyses revealed that the MA significantly increased in both groups as IOI lengthened ($p < 0.001$). In the IOIs of 1000 ms and 1500 ms, the experimental group exhibited significantly lower MA compared to the control group [1000 ms, $p = 0.044$; 1500 ms, $p < 0.001$], indicating superior synchronization performance of the experimental group in the IOIs of 1000 ms and 1500 ms (Fig 3B).

## Correction ability - Phase Correction Gain (PCG)

The PCG for the experimental group were 0.28 ± 0.08, 0.49 ± 0.14, and 0.50 ± 0.08 (IOI: 500, 1000 and 1500 ms, respectively), while for the control group, they were 0.30 ± 0.12, 0.45 ± 0.11, and 0.51 ± 0.10 (IOI: 500, 1000 and 1500 ms,

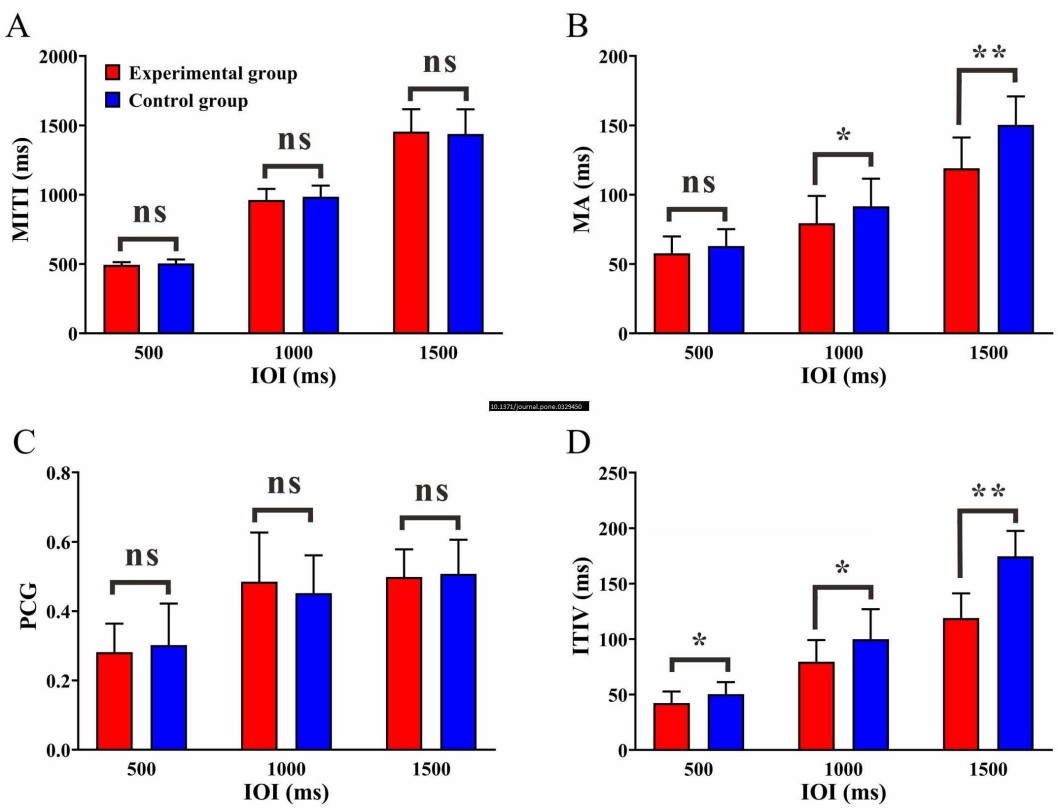

**Fig 3. Behavioral performance.** (A) The Mean intertap interval (MITI) of the two groups under three conditions. (B) The Mean Asynchrony (MA) of the two groups under three conditions. (C) The Phase Correction Gain (PCG) of the two groups under three conditions. (D) The intertap interval variability (ITIV) of the two groups under three conditions. Note: Error bars represent standard error; * designates $p < 0.05$;** designates $p < 0.01$; ns designates not significant.

respectively). Repeated-measures ANOVA revealed no significant main effect of group [$F_{(1,43)}$ = 0.004, $p$ = 0.953, $\eta^2$ = 0.014], a significant main effect of condition [$F_{(2,86)}$ = 57.711, $p < 0.001$, $\eta^2$ = 0.573], and no significant interaction effect of group × condition [$F_{(2,86)}$ = 0.911, $p > 0.397$, $\eta^2$ = 0.025]. Further simple effects analysis showed that the PCG was significantly lower in the IOI of 500 ms compared to the IOI of 1000 ms [$p < 0.001$] and IOI of 1500 ms [$p < 0.001$] (Fig 3C).

## Stability: Intertap interval variability(ITIV)

The ITIV for the experimental group were 42.43 ± 10.33 ms, 82.39 ± 26.60 ms, and 144.04 ± 22.39 ms (IOI: 500, 1000 and 1500 ms, respectively),while for the control group, they were 50.46 ± 10.75 ms, 99.86 ± 27.10 ms and 174.64 ± 22.95 ms (IOI: 500, 1000 and 1500 ms, respectively). Repeated measures ANOVA revealed a significant main effect of group [$F_{(1,43)}$ = 17.534, $p < 0.001$, $\eta^2$ = 0.290], a significant main effect of condition [$F_{(2,86)}$ = 614.162, $p < 0.001$, $\eta^2$ = 0.967], and a significant interaction effect of group × condition [$F_{(2,86)}$ = 6.240, $p < 0.004$, $\eta^2$ = 0.229]. Further simple effects analysis showed that the ITIV increased significantly with longer IOI in both groups [$p < 0.001$]. However, the ITIV was significantly smaller in the experimental group compared to the control group under all three ITIs [500 ms, $p = 0.014$; 1000 ms, $p = 0.035$; 1500 ms, $p < 0.001$], indicating better stability in the experimental group across all conditions (Fig 3D).

To elucidate the key factors influencing synchronization performance, a Pearson correlation analysis was conducted to examine the associations between the MA, PCG, and ITIV in the two participant groups. The results revealed significant positive correlations between the MA and ITIV for both groups in the ITI of 1000 ms and 1500 ms [1000 ms: experimental group, $r = 0.649$, $p = 0.001$; control group, $r = 0.768$, $p < 0.001$; 1500 ms: experimental group, $r = 0.819$, $p < 0.001$; control group, $r = 0.799$, $p < 0.001$] (Fig 4). These findings indicate that both groups rely on stability to achieve superior task performance in the IOIs of 1000 ms and 1500 ms.

## Interpersonal brain synchronization (IBS)

The findings from the one-sample t-test analysis on the IBS are as follows: In the IOI of 500 ms, the experimental group exhibited significant IBS in CH2 [$t_{(22)}$ = 3.740, $p = 0.001$, Cohen's $d = 1.103$], while the control group showed significant IBS in CH17 [$t_{(21)}$ = 2.404, $p = 0.018$, Cohen's $d = 0.725$]. However, only the experimental group's CH2 passed the FDR correction [$p = 0.001$] (Fig 5A); In the IOI of 1000 ms, the experimental group exhibited significant IBS in CH2 [$t_{(22)}$ = 6.068, $p < 0.001$, Cohen's $d = 1.790$] and CH3 [$t_{(22)}$ = 2.537, $p = 0.019$, Cohen's $d = 0.748$], while the control group showed significant IBS in CH3 [$t_{(21)}$ = 2.849, $p = 0.010$, Cohen's $d = 0.860$] and CH13 [$t_{(21)}$ = 2.832, $p = 0.010$, Cohen's $d = 0.853$]. Nevertheless, only the experimental group's CH2 passed the FDR correction [$p < 0.001$](Fig 5B); In the IOI of 1500 ms, the experimental group exhibited significant IBS in CH2 [$t_{(22)}$ = 4.706, $p < 0.001$, Cohen's $d = 1.388$], while the control group showed significant IBS in CH17 [$t_{(21)}$ = 2.335, $p = 0.030$, Cohen's $d = 0.704$]. However, only the experimental group's CH2 passed the FDR correction [$p < 0.001$](Fig 5C). CH2 is located in the right superior frontal gyrus(rSFG), and the above results indicate task-related IBS in the rSFG of the experimental group.

The IBS in CH2 for the experimental group were 0.05 ± 0.07, 0.07 ± 0.05 and 0.05 ± 0.06 (IOI: 500, 1000 and 1500 ms, respectively), while for the control group, they were −0.05 ± 0.05, −0.00 ± 0.07 and 0.00 ± 0.04(IOI: 500, 1000 and 1500 ms, respectively). Repeated measures ANOVA revealed a significant main effect of group [$F_{(1, 43)}$ = 22.049, $p < 0.001$, $\eta^2$ = 0.339], no significant main effect of condition [$F(2, 86)$ = 0.888, $p = 0.408$, $\eta^2$ = 0.020], and no significant interaction effect of group × condition [$F_{(2, 86)}$ = 0.543, $p = 0.569$, $\eta^2$ = 0.012]. Further analysis of simple effects revealed that for all three condition typess, the IBS in the experimental group was significantly higher than that in the control group [500 ms, $p < 0.001$; 1000 ms, $p < 0.001$; 1500 ms, $p < 0.001$; FDR correction]. Thus, the rSFG of the experimental group exhibited higher IBS across all three conditions (Fig 5D).

To elucidate the inter-brain mechanisms underlying the SMS task in the experimental group, Pearson correlation analyses were conducted between IBS and behavioral measures. The results revealed significant negative correlations between IBS and MA in the IOIs of 1000 ms and 1500 ms [1000 ms: $r = −0.682$, $p < 0.001$; 1500 ms: $r = −0.697$,

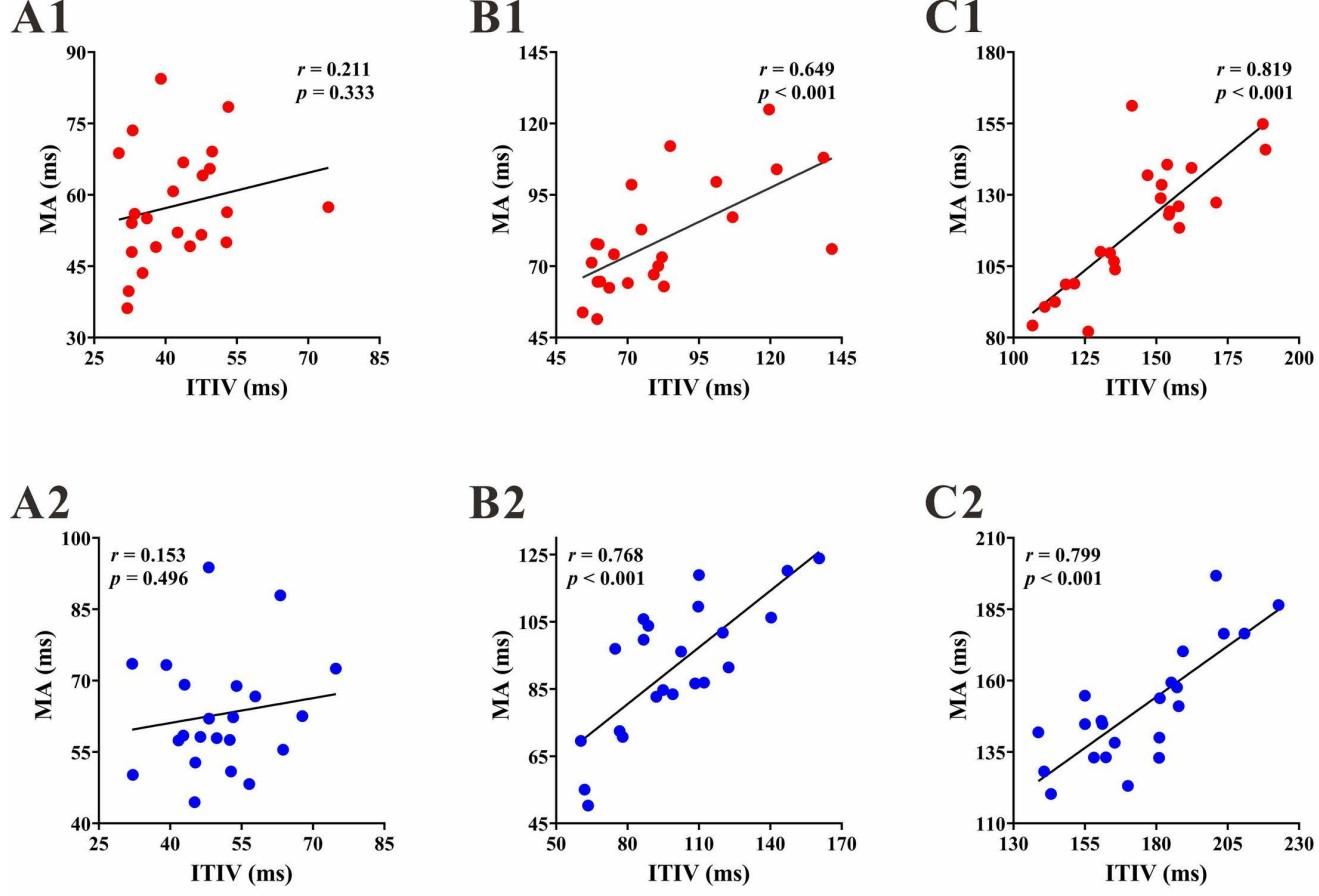

**Fig 4. Mean Asynchrony (MA) - Intertap interval variability (ITIV) correlation in two groups.** (A1) (B1) (C1) Experimental group in the ITI of 500, 1000 and 1500 ms respectively. (A2) (B2) (C2) Control group in the ITI of 500, 1000 and 1500 ms respectively.

$p < 0.001$] (Fig 6). This indicates that higher IBS is associated with better task performance. Furthermore, in all three IOIs, there were significant negative correlations between IBS and ITIV [500 ms: $r = -0.472$, $p = 0.023$; 1000 ms: $r = -0.692$, $p < 0.001$; 1500 ms: $r = -0.704$, $p < 0.001$] (Fig 7). These findings suggest that greater stability is related to higher IBS.

## Discussion

The tasks have been divided into three IOIs (500 ms, 1000 ms, 1500 ms) based on the level of difficulty. We used MA as the outcome measure of task performance, and PCG and ITIV as the process measures of task strategy. The aim was to investigate whether individuals with different aerobic fitness levels exhibit differences in SMS abilities and the potential underlying mechanisms. We employed fNIRS-based hyperscanning to explore the interpersonal brain mechanisms during the SMS task. The results showed that under a comfortable tempo condition, the experimental group did not exhibit an advantage in SMS abilities.

However, with IOIs increasing, the superiority of SMS abilities tends to be constantly improving. In terms of the components of SMS, the two groups exhibited similar phase correction abilities, and the primary reason for the difference in task performance was that the experimental group had more stable ITI.

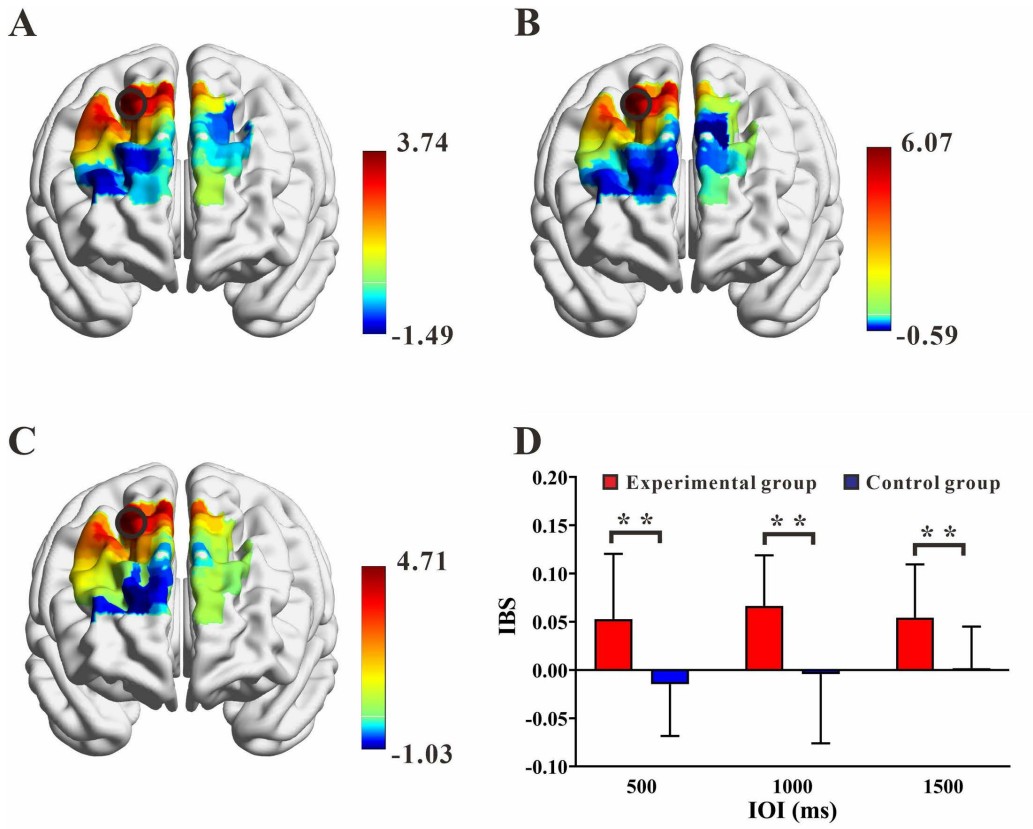

**Fig 5. Interpersonal brain synchronization (IBS).** (A) (B) (C) T-maps of IBS of experimental group in the ITI of 500, 1000 and 1500 ms respectively. (D) The IBS in channel 2 (CH2) of two groups under three conditions. Note: Error bars represent standard error; ** designates $p < 0.01$.

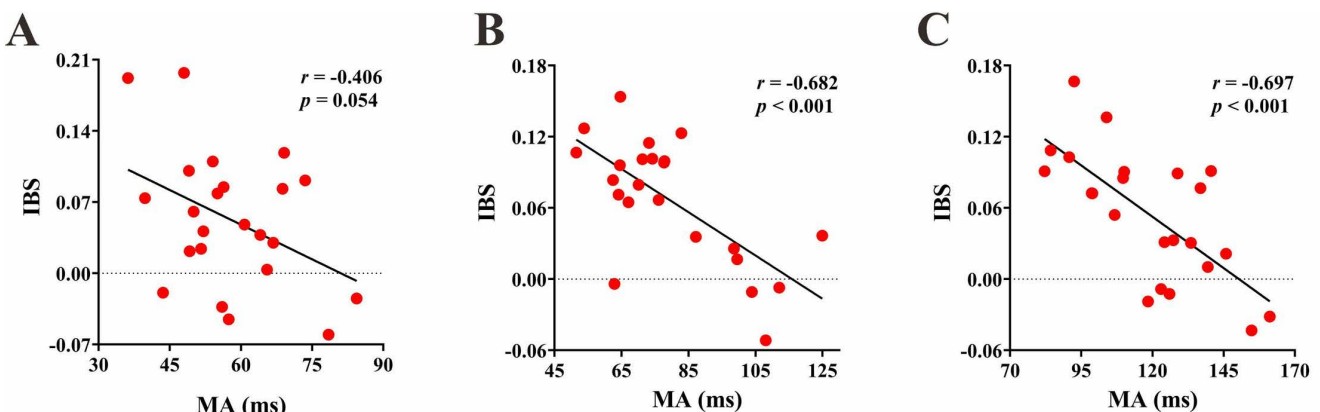

**Fig 6. Interpersonal brain synchronization (IBS) - Mean Asynchrony (MA) correlation in experimental group.** (A) (B) (C) In the ITI of 500, 1000 and 1500 ms respectively.

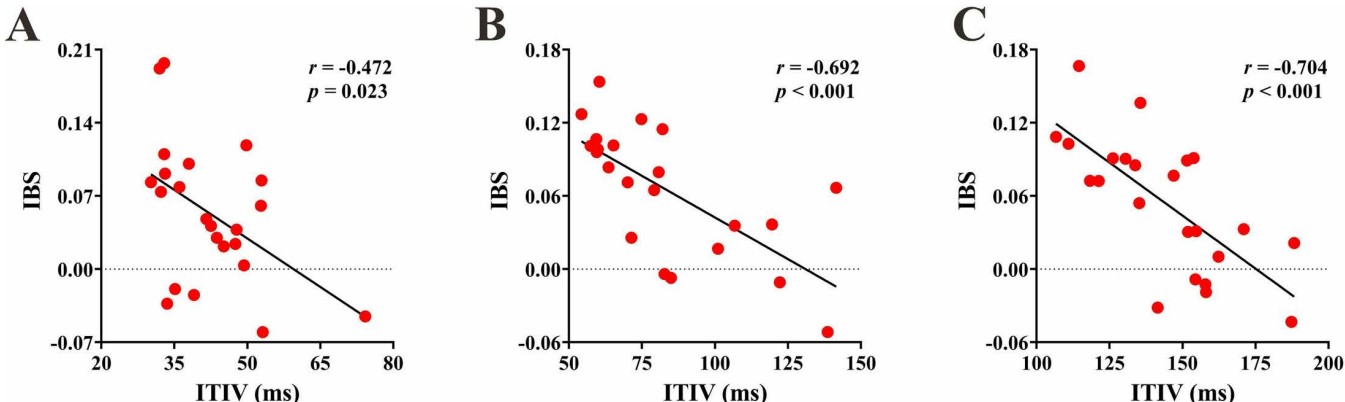

**Fig 7. Interpersonal brain synchronization (IBS)-Intertap interval variability (ITIV) correlation in experimental group.** (A) (B) (C) In the ITI of 500, 1000 and 1500 ms respectively.

It is found that the experimental group showed significantly IBS in the rSFG during the task, and this enhancement was significantly correlated with task performance as task difficulty increased at the neuroscientific level. Furthermore, the IBS in the experimental group was significantly correlated with the stability across all three conditions. Numerous studies have found a positive correlation between physical activity and cognitive abilities across different developmental stages, suggesting that higher physical activity is associated with higher cognitive abilities [44], which may be related to improved aerobic capacity [45].

The natural frequency of spontaneous human movement rhythm is approximately 2 Hz [46]. This mechanism influences various abilities, such as attention, temporal expectation, and action coordination, resulting in the smallest MA at this frequency [47]. Even individuals with specialized musical training do not exhibit differences from the general population at this frequency [18]. However, as the task frequency decreases, performance differences between groups begin to emerge. For example, musicians start to show differences from the general population in IOI of 800 ms [19], and older adults exhibit significantly larger MA compared to young controls in IOI closer to 1 s (900 ms) [48]. The results of the current study are consistent with previous behavioral findings. The experimental group did not show better performance in IOI of 500 ms, but began to demonstrate an advantage in IOI of 1000 ms. Furthermore, this advantage tended to increase as the task difficulty level rose.

Previous research suggested that the corrective ability, rather than the stability, plays a more important role in the SMS process [25]. However, the current study presents different findings. In the 500 ms IOI task, which may be considered a relatively easy rhythm preferred by most individuals, the task did not require significant cognitive resources to be completed adequately. Neither the PCG nor the ITIV of the two groups showed a significant correlation with task performance. In contrast, at the 1000 ms and 1500 ms IOIs levels, although both groups exhibited larger PCG compared to the IOI of 500 ms, these gains were not correlated with task performance. Instead, ITIV was highly correlated with task performance. This suggests that both groups employed a similar strategy, relying more on maintaining relatively stable ITI to achieve better synchronization, rather than relying heavily on the corrective ability.

The experimental group exhibited better stability performance across all three task difficulty levels, resulting in superior task outcomes. We propose two possible explanations for this finding: First, the selective facilitation hypothesis of the brain suggests that different forms of exercise have selective cognitive benefits, with aerobic activities demonstrating the most pronounced improvements in executive control related cognitive abilities [49]. Maintaining a stable ITI requires sustained attentional resources, which is a key component of executive control. Therefore, the enhanced aerobic fitness

of the experimental group may have improved their stability performance through enhanced attentional mechanisms. Second, many aerobic activities (e.g., running, walking, swimming) involve periodic movements. During these activities, individuals need to master appropriate movement frequencies and maintain a stable rhythm to optimize their physical performance. Previous research has found that long-term engagement in periodic exercise can enhance temporal perception abilities [50]. The entire exercise process is a concentrated manifestation of the interaction between perceptual and motor systems, and a process of continuously strengthening the ability to maintain a stable rhythm. The participants in the current study were all long-distance runners. Therefore, their superior stability performance may also have been shaped by the continuous reinforcement of rhythm stabilization abilities during their periodic training regimens.

The fNIRS results showed that the experimental group exhibited significantly IBS in CH2, which corresponded to the rSFG region, while the control group did not show significant IBS in any of the CHs. Previous research has found that even during unconscious synchronization processes, brain oscillations are significantly stronger compared to non-synchronized states, suggesting the existence of specific neural markers for synchronization tasks [51]. Studies have also reported enhanced IBS in the prefrontal cortex during synchronization tasks, and this IBS was positively correlated with task performance [52]. Our study further validated these previous findings in the experimental group, but the control group did not show significant results. Comparisons between basketball players and university students on cooperative tasks have similarly found that only the athlete group exhibited significantly IBS in the prefrontal cortex, along with stronger cooperative awareness [34]. There was a significant difference in IBS between the two groups in the 500 ms task. We speculate that this result may be due to the relative simplicity of the finger-tapping task, which, for ordinary individuals, elicits a lower intensity of neural impulses. Therefore, no significant IBS was observed in the control group. In contrast, higher aerobic fitness may have a beneficial effect on neural plasticity, enhancing neural impulses during the task. As a result, significant IBS was observed in the experimental group. Meta-analytic on the effects of exercise on cognition have found that regular aerobic exercise can enhance prosocial tendencies and behaviors, such as cooperation and trust [53]. In our study, better synchronization performance required close coordination between the two participants. Therefore, the experimental group's smaller MA in the IOI of 1000 ms and 1500 ms, and its strong correlation with significantly IBS in rSFG, may be attributed to their improved coordination abilities, which in turn facilitated the observed IBS. However, in the IOI of 500 ms, the experimental group's MA did not significantly correlate with the IBS. This may be because the task was closer to the natural spontaneous rhythm, and the unconscious components during the task had a greater impact on synchronization performance, which may have masked the relationship between MA and IBS.

We did not find a relationship between the experimental group's PCG and the IBS, consistent with previous suggestions that phase correction is an automatic, unconscious process that occurs within individuals [17]. So how did the experimental group's stronger coordination abilities manifest? The experimental group showed lower ITIV across all three conditions, and this variability was significantly correlated with the IBS. Previous research has suggested that making one's own behavior more predictable is a strategy for improving coordination, and reducing behavioral variability is one way to achieve this [54]. Studies have found that individuals exhibit lower response variability when engaged in joint button-pressing tasks compared to individual button-pressing, and the lower the individual's behavioral variability, the higher the coordination between the two partners [55]. Therefore, we propose that the experimental group's enhanced stability led to lower behavioral variability, providing stronger cues for their partner's next actions, which in turn improved their coordination and facilitated the observed IBS.

## Conclusion

In summary, the present study concluded:(1) Individuals with high aerobic fitness demonstrated superior SMS abilities in a seconds-scale task; (2) Both groups employed similar task strategies, but the advantage in SMS capacities demonstrated by the high-fitness individuals was attributed to their improved stability; (3) The high-fitness individuals exhibited significantly IBS in the rSFG, which was associated with their superior stability abilities.

## Supporting information

**S1. Raw and processed data files.**
(XLSX)

## Author contributions

**Conceptualization:** Haoteng Yuan, Shengyu Dai.

**Data curation:** Zixin Wang.

**Formal analysis:** Zixin Wang.

**Funding acquisition:** Shilin Wen.

**Investigation:** Zixin Wang, Haoteng Yuan, Shengyu Dai.

**Methodology:** Zixin Wang, Jianguang Cai, Hui Xu.

**Project administration:** Zixin Wang.

**Resources:** Shilin Wen.

**Software:** Zixin Wang.

**Validation:** Chengliang Wang.

**Visualization:** Liang Jiang, Xue Chen.

**Writing – original draft:** Zixin Wang, Chengliang Wang, Xianzhi Gao.

**Writing – review & editing:** Shilin Wen.

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
