## [Decision Letter · Decision Letter 0]

PONE-D-24-44814Interpersonal brain synchronization during sensorimotor synchronization in people with different aerobic fitness levels: A fNIRS-Based hyperscanning StudyPLOS ONE

Dear Dr. Wen,

Thank you for submitting your manuscript to PLOS ONE. After careful consideration, we feel that it has merit but does not fully meet PLOS ONE’s publication criteria as it currently stands. Therefore, we invite you to submit a revised version of the manuscript that addresses the points raised during the review process.

**ACADEMIC EDITOR:**

Sorry for the long delay.  It has been difficult to secure reviewers. After careful consideration, we feel that it has merit but does not fully meet PLOS ONE’s publication criteria as it currently stands. Therefore, we invite you to submit a revised version of the manuscript that addresses the points raised during the review process. Please address reviewers comments such as justification of participants groups, terminologies used, many issues in data analysis. 

We look forward to receiving your revised manuscript.

Kind regards,

Hong-jin Sun, Ph.D.

Academic Editor

PLOS ONE

Journal Requirements:

For additional information about PLOS ONE ethical requirements for human subjects research, please refer to http://journals.plos.org/plosone/s/submission-guidelines#loc-human-subjects-research .

 [Ministry of Education, Humanities and Social Sciences project (No. 22YJA890028]. 

5. In the online submission form, you indicated that [All data can be obtained from the corresponding author.

Capital University of Physical Education And Sports Beijing Institute of Technology,

11 North Third Ring Road West

Beijing 100191, People’s Republic of China

Email: wenshilin@cupes.edu.cn].

Reviewers' comments:

Reviewer's Responses to Questions

**Comments to the Author**

1. Is the manuscript technically sound, and do the data support the conclusions?

Reviewer #1: Partly

Reviewer #2: Yes

Reviewer #3: Yes

2. Has the statistical analysis been performed appropriately and rigorously? 

Reviewer #1: Yes

Reviewer #2: Yes

Reviewer #3: No

3. Have the authors made all data underlying the findings in their manuscript fully available?

Reviewer #1: Yes

Reviewer #2: Yes

Reviewer #3: Yes

4. Is the manuscript presented in an intelligible fashion and written in standard English?

Reviewer #1: Yes

Reviewer #2: Yes

Reviewer #3: Yes

5. Review Comments to the Author

Reviewer #1: This study examines the differences in inter-brain synchrony signals during a sensorimotor synchronization (SMS) task in individuals with varying levels of aerobic fitness. The study recruited long-distance runners and non-athletes as the experimental and control groups, respectively, and recorded their brain activity while performing a finger-tapping synchronization task using functional near-infrared spectroscopy (fNIRS). The results revealed that the experimental group, with higher aerobic fitness, demonstrated superior synchronization abilities and greater inter-brain synchrony, particularly in the right prefrontal cortex. These findings provide new evidence for the relationship between aerobic exercise, cognitive function, and the underlying neural mechanisms. However, several aspects of the study require further clarification.

Introduction

1. The term “sensorimotor synchronization” is misspelled (e.g., “Sensoriotor synchronization”). Please review the manuscript for consistency and correct this error throughout.

2. There is a lack of discussion on the relationship between aerobic exercise and SMS. How did the authors arrive at the conclusion “it is still unclear whether good aerobic fitness is associated with enhanced SMS ability and its potential mechanisms”? Why were these two variables linked? Is there a debate in the existing literature on this topic, or are theoretical concepts in this area yet to be verified? Please consider expanding on this point.

3. The concept of ROI (Region of Interest) is not clearly defined. The manuscript only discusses the relationship between synchronized movement and the prefrontal or sensory-motor cortex, but it does not explain whether aerobic exercise and different rhythms also influence the prefrontal cortex. Further discussion on this is recommended.

4. The study hypothesis is not included. Please add a research hypothesis grounded in theoretical or empirical evidence to address the research question.

Methods

1. Why were long-distance runners specifically chosen for this study? What role do these subjects play in addressing the research question? Could other types of aerobic exercise groups also be applicable?

2. Please clarify which irrelevant variables were controlled for in both the experimental and control groups. Did the control group assess aerobic fitness levels, and were subjects with expertise in rhythm control (e.g., pianists) or extensive aerobic exercise experience (e.g., regular gym-goers) excluded?

3. The rationale for using 1000m run time as an indicator of aerobic fitness is unclear. Please provide supporting references.

4. Please verify the consistency of channel placements across subjects. What is the variability in channel location data (MNI coordinates) between the two groups, and how do these differ?

5. In the data analysis section, the manuscript mentions that "HbO is more sensitive to blood flow changes." Please include supporting references for this claim.

6. In Figure 2, is the data presented from a single subject or is it the average of all subjects?

Results

1. Please include an analysis of the validity of the subjects’ behavioral responses, specifically comparing the MITI between the two groups and their corresponding control rhythms.

2. Is there a correlation between the aerobic fitness level of the experimental group and IBS?

3. Does the relationship between MA/ITIV and IBS differ between the experimental and control groups?

Discussion

1. Under the 500ms condition, there were no differences in MA between the two groups, but significant differences in inter-brain synchrony were observed. What explains this discrepancy between the behavioral and brain activity results? Please discuss in detail.

Reviewer #2: The manuscript reports findings regarding the beneficial effects of good physical fitness on the ability to perform synchronized actions.

The study presents the results of original research.

Results reported have not been published elsewhere.

Experiments, statistics, and other analyses are performed to a high technical standard and are described in sufficient detail.

Conclusions are presented in an appropriate fashion and are supported by the data.

The article is presented in an intelligible fashion and is written in standard English.

The research meets all applicable standards for the ethics of experimentation and research integrity.

The article adheres to appropriate reporting guidelines and community standards for data availability.

There is only one point that the authors should consider

The manuscript title names as "interpersonal synchronization" the correspondence between physiological variables of two individuals responding to the same external sytimuli. This is not interpersonal synchronization,that is usually referred to the similar behavioral and physiological activities observable in two persons looking at each other and reflecting each other.

Thus, I suggest to change the title, erasing the "interpersonal synchronization" , and focus the descrpition of results and discussion on fi effect of fitness on the individual ability to follow the external stimuli

Reviewer #3: General comments.

This is a paper studying the influence of aerobic fitness on sensorimotor synchronization and interpersonal brain synchronization.

In general, the experimental methodology used was appropriate. Two experimental groups, differing in the degree of aerobic fitness were compared. Methodological technics using on one side, behavioral evaluation parameters, and on the other functional near-infrared spectroscopy were consistent to determine the possible changes.

Although the two functional properties addressed in the paper (Interpersonal Brain Synchronization and Sensorimotor Synchronization), are explicitly defined in the text, some confusing research aim is detected during the exposition of the work, were the reader navigates between the Interpersonal Brain Synchronization or the Sensorimotor Synchronization, as the main point of study.

Specific comments.

1) Authors use to much abbreviations, which does not facilitate the understanding of the text. Perhaps it should be convenient and if authors agree, the use of a table with all the abbreviations used in the text at the end of the manuscript. This is one of the difficulties in following authors ‘arguments.

2) Discrepant definitions between declarations of variables in Methods and their presentations in the figures (“Synchrony Performance, Average asynchrony identified as “AS”, pp. 16, and in Figure 4, as “MA”; Inter-Brain synchronization identified as “IBS”, pp. 9, and Interpersonal Brain Synchronization with the same abbreviation [IBS] Figure 6). Declarations of variables with the same meaning or the same abbreviations should be consistent along the paper to avoid confusing the reader.

3) Paper is focalized into Interpersonal Brain Synchronization and Sensorimotor Synchronization. However, no explicit declaration about how these variables are experimentally measured and its meaning when values of variables changes in its range is given in Material and Methods. It will be very helpful to general readers if authors give an additional complementary explanation.

4) There is some technically problem with the correlation-regression analysis performed in figures 4-7.

In page 8, it is declared that MA in the IOI= 1000 ms is 79.50 ± 19.1. This means that for 79.5 ms the most higher value is 98.6 ms, and the most lower value is 60.4 ms. Statistically, this range represent one population with a mean value of 79.50 ms. In Figure 6 B, as an example, the range of values for the independent variable (MA), 60.4 -98.6 ms, represent just only one value for the dependent variable IBS. Thus, the cluster of points of IBS between 60.4 – 98.6 ms are not really different covariant values of MA. They represent only an estimate of the covariant IBS, when MA = 79.50. This is so, because authors decided to use the statistical Model 2 for the regression analysis. My suggestion is that the correlation-regression analysis should be eliminated, since authors have enough evidence supporting the hypothesis that the condition of aerobic fitness is influencing the parameters under study. A similar comment for the rest is valid.

6. PLOS authors have the option to publish the peer review history of their article (what does this mean? ). If published, this will include your full peer review and any attached files.

**Do you want your identity to be public for this peer review?** For information about this choice, including consent withdrawal, please see our Privacy Policy .

Reviewer #1: **Yes: ** Qihan Zhang

Reviewer #2: **Yes: ** Enrica Laura SANTARCANGELO

Reviewer #3: **Yes: ** Edgardo O. Alvarez

---

## [Author Response · Author response to Decision Letter 1]

27 Jun 2025

Cover letter

Dear Dr. Sun,

On behalf of all the contributing authors, I would like to express our sincere appreciation of your letter and reviewers’ constructive comments concerning our article entitled“Interpersonal brain synchronization during sensorimotor synchronization in people with different aerobic fitness levels: A fNIRS-Based hyperscanning study”(Manuscript ID PONE-D-24-44814).

We have considered the comments very carefully and have revised the paper accordingly.These comments are all valuable and helpful for improving our article. Point-by-point responses to the comments from the editor and the three reviewers can be found in the Responses to the Editor and Reviewers.

We confirm that all authors have reviewed and approved the content of the manuscript and agree to its submission to your journal. Information regarding funding sources and ethical statements has been disclosed in the manuscript as required. In addition, the funders were involved in the experimental design, data analysis, and the writing and revision of the manuscript.

Thanks again for your consideration of publishing our manuscript in your journal. We are looking forward to hearing from you soon.

Please let me know if any additional information is required.

Sincerely,

Shilin Wen, Ph.D., Professor

Capital University of Physical Education and Sports

Beijing 100091, People’s Republic of China

Email: 313811819@qq.com

Phone: +86-18001090696

Response to the editor’s and reviewers’ comments

Dear Editor and Reviewers

We sincerely appreciate your thorough review of our manuscript. We are grateful for your constructive comments and suggestions, which have greatly helped us improve the quality of our manuscript.

We have carefully considered all the comments and have revised the manuscript accordingly. The revised content has been marked in red color.

Below, we provide a detailed point-by-point response to each comment.

Journal Requirements:

1.Please ensure that your manuscript meets PLOS ONE's style requirements, including those for file naming.

Response: Thank you for your reminder.

We have revised the format of the manuscript according to the requirements of the journal.

2.PLOS requires an ORCID iD for the corresponding author in Editorial Manager on papers submitted after December 6th, 2016. Please ensure that you have an ORCID iD and that it is validated in Editorial Manager. To do this, go to ‘Update my Information’ (in the upper left-hand corner of the main menu), and click on the Fetch/Validate link next to the ORCID field. This will take you to the ORCID site and allow you to create a new iD or authenticate a pre-existing iD in Editorial Manager..

Response: Thank you for your reminder.

We have verified the ORCID iD in the Editorial Manager system.

3.Please provide additional details regarding participant consent. In the ethics statement in the Methods and online submission information, please ensure that you have specified what type you obtained (for instance, written or verbal, and if verbal, how it was documented and witnessed). If your study included minors, state whether you obtained consent from parents or guardians. If the need for consent was waived by the ethics committee, please include this information.

Response: We have added an Ethics Statement in the Materials and Methods section of the manuscript. The specific content is as follows:

"This study was performed in accordance with the ethical standards specified in the latest Helsinki Declaration and was approved by the ethics committee of Capital University of Physical Education and Sports (2021A21). All participants signed the informed consent before the experiment and received certain remuneration after completing the experimental task."

[Ministry of Education, Humanities and Social Sciences project (No. 22YJA890028].

Response: Thank you for your reminder.

The funders were involved in the experimental design, data analysis, and the writing and revision of the manuscript. We have included this information in the cover letter.

5.All PLOS journals now require all data underlying the findings described in their manuscript to be freely available to other researchers, either 1. In a public repository, 2. Within the manuscript itself, or 3. Uploaded as supplementary information.

Response: Thank you for your reminder.

We have uploaded the data as supplementary information and have indicated this in the revised manuscript.

Reviewer #1:

Introduction

1.The term “sensorimotor synchronization” is misspelled (e.g., “Sensoriotor synchronization”). Please review the manuscript for consistency and correct this error throughout.

Response: Thank you for your helpful suggestions.

We have thoroughly checked and corrected all the spelling mistakes in the manuscript.

2.There is a lack of discussion on the relationship between aerobic exercise and SMS. How did the authors arrive at the conclusion “it is still unclear whether good aerobic fitness is associated with enhanced SMS ability and its potential mechanisms”? Why were these two variables linked? Is there a debate in the existing literature on this topic, or are theoretical concepts in this area yet to be verified? Please consider expanding on this point.

Response: Thank you for your helpful comments and suggestions.

We reviewed the published literature and did not find studies that specifically investigated the relationship between aerobic fitness and SMS ability. Therefore, we proposed that "it is still unclear whether good aerobic fitness is associated with enhanced SMS ability and its potential mechanisms." In addition, in the second and third paragraphs of the introduction, we explain how these two variables are linked. In the second paragraph of the introduction, we discuss the possible relationship between SMS and physical movement, providing examples of sports activities that can enhance SMS ability. In the third paragraph, we describe how aerobic exercise can improve brain structure and function and promote cognitive development, and note that it remains unclear whether aerobic exercise can improve SMS as a cognitive function.

3.The concept of ROI (Region of Interest) is not clearly defined. The manuscript only discusses the relationship between synchronized movement and the prefrontal or sensory-motor cortex, but it does not explain whether aerobic exercise and different rhythms also influence the prefrontal cortex. Further discussion on this is recommended.

Response: Thank you for your helpful comments and suggestions.

In the revised manuscript, we defined the concept of ROI in the Methods—Data Collection section as follows:

The prefrontal cortex was selected as the region of interest (ROI) for this study.

Since our study employs a hyperscanning paradigm for simultaneous multi-brain data collection, there are only a few hundred published experimental studies overall, and to date, no research has been found exploring the relationship between aerobic exercise, different rhythms, and the prefrontal cortex. So far, we have only identified two hyperscanning studies related to physical activities and the prefrontal cortex, which we introduced in the revised manuscript as follows:

Meanwhile, hyperscanning studies related to physical activities have found that both cycling and basketball are associated with enhanced inter-brain synchronization (IBS) in the prefrontal cortex, as well as improvements in specific task performance.

Regarding the variable of different rhythms, we provided the following introduction in the fourth paragraph of the Introduction:

In laboratory research on SMS, information processing methods are commonly employed, with finger-tapping being a typical task where individuals strive to synchronize their finger taps with an external rhythm. It is found that the performance of SMS tasks is directly influenced by the sequence interonset interval (IOI), with the optimal synchronization frequency occurring at approximately 2 Hz (around 400 ms–600 ms). At this frequency, performance differences among different populations are relatively small. However, as IOI increases, task performance declines and individual differences become more pronounced.

4.The study hypothesis is not included. Please add a research hypothesis grounded in theoretical or empirical evidence to address the research question.

Response: Thank you for your helpful comments and suggestions.

We have added the following hypothesis: Based on the beneficial effects of aerobic exercise on cognitive function and the enhancement of IBS in the prefrontal cortex induced by physical activity, the present study proposes the following hypothesis: Individuals with higher levels of aerobic fitness may exhibit superior SMS abilities, an advantage that becomes more pronounced as the IOI increases, possibly due to aerobic exercise enhancing task performance by facilitating IBS in the prefrontal cortex.

Methods

1.Why were long-distance runners specifically chosen for this study? What role do these subjects play in addressing the research question? Could other types of aerobic exercise groups also be applicable?

Response� Thanks for your comment.

There are two main reasons for selecting long-distance runners as participants in our study. First, from a training perspective, long-distance running is one of the most representative forms of aerobic exercise, which closely matches the aerobic fitness variable we are investigating. Second, our experiment was conducted during the COVID-19 pandemic, when it was difficult for other populations to engage in regular and systematic exercise. In contrast, the long-distance runners received organized training provided by the university, which helped them maintain a high level of aerobic fitness during the experimental period. As a result, these participants served as an effective means of controlling variables in addressing the research questions. Since most types of aerobic exercise possess periodic or rhythmic characteristics, we believe that our findings are applicable to most populations engaged in aerobic exercise.

2.Please clarify which irrelevant variables were controlled for in both the experimental and control groups. Did the control group assess aerobic fitness levels, and were subjects with expertise in rhythm control (e.g., pianists) or extensive aerobic exercise experience (e.g., regular gym-goers) excluded?

Response: Thank you for your helpful suggestions.

Except for the difference in exercise duration between the two groups, we tried to control for the consistency of other variables that might affect the experimental results. These included participants’ handedness, the level of familiarity within each matched pair, and musical instrument learning background. In addition, we ensured that members of the control group had less than one hour of exercise per week over the past three months, with no systematic gym training experience. In response to your comments, we have added the following content to the Participants section in the revised manuscript: None of the participants had received systematic musical instrument training. Members of the experimental group engaged in an average of no less than 6 hours of aerobic exercise per week in the past three months, while members of the control group self-reported an average of no more than 1 hour of exercise per week during the same period and had no history of systematic gym workouts.

3.The rationale for using 1000m run time as an indicator of aerobic fitness is unclear. Please provide supporting references.

Response: Thank you for your suggestions.

In the manuscript, we included two references after the sentence: “One week prior to the experiment, the participants' 1000-meter running times were recorded as a measure of their aerobic fitness level.” The first reference provides the following statement: “Field tests such as the 12-min run, 1.5-mile (2.4 km) run, and 1000-m run are frequently used to estimate cardiorespiratory fitness”1. The second reference supports the reliability of the 1000-meter run as an evaluation tool for aerobic fitness among elite soccer players2.

A maximal multistage 20-m shuttle run test to predict VO2 max

The concurrent validity and between-session reliability of a 1000m time trial The concurrent validity and between-session reliability of a 1000m time trial for the assessment of aerobic fitness in elite development soccer players

4.Please verify the consistency of channel placements across subjects. What is the variability in channel location data (MNI coordinates) between the two groups, and how do these differ?

Response: Thank you for your helpful suggestions.

We calculated the average MNI coordinates and corresponding Brodmann areas for both the experimental group and the control group. We found that the coordinates of each channel were essentially the same between the two groups, and all 17 channels corresponded to the same Brodmann areas. According to your suggestion, we have revised Table 1 in the manuscript. The revised table is shown below.

Table 1. The coordinates in MNI space and corresponding neuroanatomical labels for channels in the ROI.

Channel MNI coordinates(E/C) Brodman area

X Y Z

1 27/28 25/24 61/62 Dorsolateral prefrontal cortex

2 5/6 27/26 64/65 Dorsolateral prefrontal cortex

3 -17/-17 28/28 63/63 Dorsolateral prefrontal cortex

4 41/40 32/31 48/48 Dorsolateral prefrontal cortex

5 18/19 36/36 58/58 Dorsolateral prefrontal cortex

6 -7/-6 40/40 57/57 Dorsolateral prefrontal cortex

7 -26/-26 36/36 53/51 Dorsolateral prefrontal cortex

8 30/31 44/44 45/44 Dorsolateral prefrontal cortex

9 10/11 48/49 52/51 Pre-Motor and Supplementary Motor Cortex

10 -14/-13 48/49 49/49 Includes Frontal eye fields

11 46/46 49/48 26/27 Includes Frontal eye fields

12 23/24 56/56 38/39 Pre-Motor and Supplementary Motor Cortex

13 0/0 56/56 39/39 Pre-Motor and Supplementary Motor Cortex

14 -23/-23 56/57 36/36 Pre-Motor and Supplementary Motor Cortex

15 38/37 60/60 20/21 Pre-Motor and Supplementary Motor Cortex

16 14/15 67/66 29/28 Includes Frontal eye fields

17 -10/-10 66/66 28/28 Includes Frontal eye fields

Note�E represents the experimental group, and C represents the control group.

5.In the data analysis section, the manuscript mentions that "HbO is more sensitive to blood flow changes." Please include supporting references for this claim.

Response: Thank you for your helpful suggestions.

We cited two references after this sentence to support this point3,4.

6. In Figure 2, is the data presented from a single subject or is it the average of all subjects?

Response: Thank you for your helpful suggestions.

This figure shows the HbO data from channel 2 of a paired participant. We described it in the manuscript as follows: "The coherence based on HbO signal from channel 2 (CH2) in a representative pair of the experimental group."

Results

1.Please include an analysis of the validity of the subjects’ behavioral responses, specifically comparing the MITI between the two groups and their corresponding control rhythms.

Response: Thank you for your helpful comments.

We provided the following introduction to the MITI in the Behavioral Data Processing and Analysis section of the Methods: Mean intertap interval(MITI): The MITI is used to assess whether the two groups of participants performed the task under the same conditions. It is obtained by calculating the mean interval between key presses for each participant within each trial. Previous research has found a strong correlation between the MITI and task performance. If there is a significant difference between the two groups, it indicates inconsistent task conditions between the groups, which can directly impact other experimental results.

In the

---

## [Decision Letter · Decision Letter 1]

PONE-D-24-44814R1Interpersonal brain synchronization during sensorimotor synchronization in people with different aerobic fitness levels: A fNIRS-based hyperscanning studyPLOS ONE

Dear Dr. Wen,

Thank you for submitting your manuscript to PLOS ONE. After careful consideration, we feel that it has merit but does not fully meet PLOS ONE’s publication criteria as it currently stands. Therefore, we invite you to submit a revised version of the manuscript that addresses the points raised during the review process.  Please refer to Reviewer 1's comments.

We look forward to receiving your revised manuscript.

Kind regards,

Hong-jin Sun, Ph.D.

Academic Editor

PLOS ONE

Journal Requirements:

Reviewers' comments:

Reviewer's Responses to Questions

**Comments to the Author**

1. If the authors have adequately addressed your comments raised in a previous round of review and you feel that this manuscript is now acceptable for publication, you may indicate that here to bypass the “Comments to the Author” section, enter your conflict of interest statement in the “Confidential to Editor” section, and submit your "Accept" recommendation.

Reviewer #1: (No Response)

Reviewer #2: All comments have been addressed

Reviewer #3: (No Response)

2. Is the manuscript technically sound, and do the data support the conclusions?

Reviewer #1: (No Response)

Reviewer #2: Yes

Reviewer #3: (No Response)

3. Has the statistical analysis been performed appropriately and rigorously? 

Reviewer #1: (No Response)

Reviewer #2: Yes

Reviewer #3: (No Response)

4. Have the authors made all data underlying the findings in their manuscript fully available?

Reviewer #1: (No Response)

Reviewer #2: Yes

Reviewer #3: (No Response)

5. Is the manuscript presented in an intelligible fashion and written in standard English?

Reviewer #1: (No Response)

Reviewer #2: Yes

Reviewer #3: (No Response)

6. Review Comments to the Author

Reviewer #1: Regarding the introduction section, it seems the author may have misunderstood my original question.

The second paragraph of the introduction emphasizes two main points. First, that sensorimotor synchronization (SMS) training can enhance motor performance. Second, that motor experience influences SMS. Based on these two points, the author concludes that there is a bidirectional relationship between SMS and physical activity. However, the focus of the present study is aerobic fitness. What is the relationship between aerobic fitness and physical activity? Can aerobic fitness be equated with physical activity? The core concepts in these two paragraphs are inconsistent.

Moreover, based on the main research content of this study, the independent variables are aerobic fitness level and IOI (inter-onset interval, i.e., rhythm), while the dependent variables include measures of synchronization performance and neural signals reflecting inter-brain synchronization. Therefore, the introduction should primarily address how the independent variables influence the dependent variables and their possible neural underpinnings—in other words, how aerobic fitness and rhythm affect synchronization, particularly the underlying inter-brain mechanisms.

Specifically, the emphasis should be on how aerobic fitness modulates SMS under different rhythmic conditions. For example, rhythm affects the difficulty of synchronization. According to the forward prediction model, individuals form predictions in advance about the sensory consequences of their actions and compare these predictions with actual sensory feedback to make corrections. Aerobic exercise continuously trains this “prediction–execution–feedback–adjustment” loop, thereby enhancing motor performance. Thus, the higher the individual’s aerobic fitness level, the more robust their forward prediction model becomes, leading to better motor performance across varying levels of difficulty.

Additionally, based on dynamic systems theory, aerobic exercise strengthens the dynamic regulation of perception–action–environment coupling, which in turn enhances synchronization performance.

Furthermore, the discussion of neuroimaging in the introduction is insufficient. For instance, in the third paragraph, what specific brain regions are affected by aerobic exercise-induced plasticity? What roles do these regions play in SMS? The second-to-last paragraph of the introduction only mentions that the prefrontal cortex is involved in SMS, but what is its relationship with aerobic exercise?

A more coherent line of reasoning might be: aerobic exercise enhances the forward prediction model, which is associated with the prefrontal cortex and premotor cortex. SMS under different IOI conditions involves the prefrontal cortex and sensorimotor cortex. Both mechanisms implicate the prefrontal and sensorimotor cortices. The more synchronized individuals’ behavioral performance is, the more accurate their prediction of each other’s movements, which is reflected in more synchronized neural activity—in other words, higher inter-brain synchrony.

Reviewer #2: My points have been addressed. The study is methodologically sound and the results are clearly reported and discussed

Reviewer #3: (No Response)

7. PLOS authors have the option to publish the peer review history of their article (what does this mean? ). If published, this will include your full peer review and any attached files.

**Do you want your identity to be public for this peer review?** For information about this choice, including consent withdrawal, please see our Privacy Policy .

Reviewer #1: No

Reviewer #2: **Yes: ** Enrica Laura SANTARCANGELO

Reviewer #3: **Yes: ** Edgardo O. Alvarez

---

## [Author Response · Author response to Decision Letter 2]

16 Jul 2025

Dear Reviewers

We sincerely appreciate your thorough review of our manuscript. We are grateful for your constructive comments and suggestions, which have greatly helped us improve the quality of our manuscript.

We have carefully considered all the comments and have revised the manuscript accordingly. The revised content has been marked in red color.

Below is our detailed response to your comments.

Reviewer #1:

Reviewer #1: Regarding the introduction section, it seems the author may have misunderstood my original question.

The second paragraph of the introduction emphasizes two main points. First, that sensorimotor synchronization (SMS) training can enhance motor performance. Second, that motor experience influences SMS. Based on these two points, the author concludes that there is a bidirectional relationship between SMS and physical activity. However, the focus of the present study is aerobic fitness. What is the relationship between aerobic fitness and physical activity? Can aerobic fitness be equated with physical activity? The core concepts in these two paragraphs are inconsistent.

Moreover, based on the main research content of this study, the independent variables are aerobic fitness level and IOI (inter-onset interval, i.e., rhythm), while the dependent variables include measures of synchronization performance and neural signals reflecting inter-brain synchronization. Therefore, the introduction should primarily address how the independent variables influence the dependent variables and their possible neural underpinnings—in other words, how aerobic fitness and rhythm affect synchronization, particularly the underlying inter-brain mechanisms.

Specifically, the emphasis should be on how aerobic fitness modulates SMS under different rhythmic conditions. For example, rhythm affects the difficulty of synchronization. According to the forward prediction model, individuals form predictions in advance about the sensory consequences of their actions and compare these predictions with actual sensory feedback to make corrections. Aerobic exercise continuously trains this “prediction–execution–feedback–adjustment” loop, thereby enhancing motor performance. Thus, the higher the individual’s aerobic fitness level, the more robust their forward prediction model becomes, leading to better motor performance across varying levels of difficulty.

Additionally, based on dynamic systems theory, aerobic exercise strengthens the dynamic regulation of perception–action–environment coupling, which in turn enhances synchronization performance.

Furthermore, the discussion of neuroimaging in the introduction is insufficient. For instance, in the third paragraph, what specific brain regions are affected by aerobic exercise-induced plasticity? What roles do these regions play in SMS? The second-to-last paragraph of the introduction only mentions that the prefrontal cortex is involved in SMS, but what is its relationship with aerobic exercise?

A more coherent line of reasoning might be: aerobic exercise enhances the forward prediction model, which is associated with the prefrontal cortex and premotor cortex. SMS under different IOI conditions involves the prefrontal cortex and sensorimotor cortex. Both mechanisms implicate the prefrontal and sensorimotor cortices. The more synchronized individuals’ behavioral performance is, the more accurate their prediction of each other’s movements, which is reflected in more synchronized neural activity—in other words, higher inter-brain synchrony.

Response: Thank you for your helpful suggestions.

We have carefully revised the introduction section of the manuscript in accordance with your suggestions. The specific modifications are as follows:

(1) We have removed the original second paragraph from the manuscript.

(2) In accordance with your suggestions, we have incorporated a discussion of neuroimaging findings into the revised second paragraph (which corresponds to the third paragraph in the original manuscript). Specifically, we have included examples illustrating the enhancing effects of aerobic exercise on the prefrontal cortex, and further clarified the role of the prefrontal cortex in SMS tasks. The revised paragraph now proceeds as follows: it first introduces the concept of aerobic fitness and its facilitating effects on various cognitive functions. It then explains, from a neuroimaging perspective, that aerobic exercise increases the activation level of the prefrontal cortex. Next, it emphasizes the critical role of the prefrontal cortex in SMS tasks. Finally, it points out the current limitation in the literature, namely the lack of studies directly examining the relationship between aerobic fitness and SMS ability. The revised content is as follows:

Aerobic fitness refers to the capacity to sustain muscular power or speed over extended periods, and is also commonly termed cardiovascular fitness. It is well established that superior aerobic fitness can be achieved through regular physical exercise. A substantial body of research has demonstrated that consistent engagement in physical activity, or possessing a high level of aerobic fitness, not only contributes to overall physical health but also increases the production of beneficial neurochemicals during exercise. This, in turn, enhances neural circuit efficiency, induces structural and functional changes in the brain, and yields long-lasting, positive effects on a range of cognitive functions. Evidence from neuroimaging studies indicates that a key brain region involved in these effects is the prefrontal cortex. For example, For example, studies have found that aerobic exercise can increase the volume and density of the prefrontal cortex, thereby enhancing memory function. In another study, children aged 8–9 years engaged in one hour of physical activity daily after school for nine months. Functional MRI was used to monitor cerebral blood flow during a flanker task that assessed attention and inhibitory control. Results revealed that these children exhibited behavioral performance and prefrontal activation patterns comparable to those of college students. In the context of SMS tasks, the prefrontal cortex operates as part of the “cognitively controlled timing” system and plays a critical role in task execution. However, it remains unclear whether optimal aerobic fitness can enhance SMS abilities, and the underlying neural mechanisms require further investigation.

(3) In accordance with your suggestions, we have revised the third paragraph (originally the fourth paragraph) to include a discussion of how aerobic exercise modulates the “prediction–execution–feedback–adjustment” loop within the forward prediction model, as well as the way in which aerobic exercise enhances the dynamic regulation of perception–action–environment coupling from the perspective of dynamic systems theory. In the revised manuscript, we have established a clearer connection between aerobic fitness and SMS performance under varying IOI conditions. The revised content is as follows:

In laboratory research on SMS, information processing methods are commonly employed, with the finger-tapping being a typical task where individuals strive to synchronize their finger taps with an external rhythm. It is found that the performance of SMS tasks is directly influenced by the sequence interonset interval (IOI) , with the optimal synchronization frequency occurring at approximately 2 Hz (around 400 ms-600 ms). At this frequency, performance differences among different populations are relatively small. However, with IOI extending, task performance declined and difference among individual increased.The forward prediction model offers a plausible explanation for these findings. According to this model, individuals generate predictions in advance regarding the sensory consequences of their actions and subsequently compare these predictions with actual sensory feedback to make corrections—this forms the so-called "prediction– execution– feedback– adjustment" loop. The key components of this loop are prediction and adjustment. As the IOI increases, the difficulty in generating accurate predictions becomes substantially greater, thereby leading to increased individual variability. Aerobic exercise has been shown to promote structural and functional plasticity in brain regions intimately involved in motor control, such as the prefrontal cortex. This enhancement boosts the efficiency of information integration and processing within these regions, thereby strengthening the predictive and adaptive elements of the aforementioned loop. Furthermore, in line with dynamic systems theory, aerobic exercise augments the dynamic regulation of the perception– action– environment coupling. Thus, we hypothesize that individuals with higher levels of aerobic fitness may demonstrate superior performance in SMS tasks.

---

## [Editor Report · Decision Letter 2]

Interpersonal brain synchronization during sensorimotor synchronization in people with different aerobic fitness levels: A fNIRS-based hyperscanning study

PONE-D-24-44814R2

Dear Dr. Wen,

We’re pleased to inform you that your manuscript has been judged scientifically suitable for publication and will be formally accepted for publication once it meets all outstanding technical requirements.

Kind regards,

Hong-jin Sun, Ph.D.

Academic Editor

PLOS ONE